# Path Planning Optimization for Driverless Vehicle in Parallel Parking Integrating Radial Basis Function Neural Network

**Leiyan Yu \*, Xianyu Wang, Zeyu Hou, Zaiyou Du, Yufeng Zeng and Zhaoyang Mu**

School of Mechanical and Electronic Engineering, China University of Petroleum, Qingdao 266580, China; Z21040031@s.upc.edu.cn (X.W.); z20040090@s.upc.edu.cn (Z.H.); Z20040007@s.upc.edu.cn (Z.D.); 1804030110@s.upc.edu.cn (Y.Z.); s21040030@s.upc.edu.cn (Z.M.)
\* Correspondence: leiyanyu@upc.edu.cn; Tel.: +86-182-6663-9167

**Abstract:** To optimize performances such as continuous curvature, safety, and satisfying curvature constraints of the initial planning path for driverless vehicles in parallel parking, a novel method is proposed to train control points of the Bézier curve using the radial basis function neural network method. Firstly, the composition and working process of an autonomous parking system are analyzed. An experiment concerning parking space detection is conducted using an Arduino intelligent minicar with ultrasonic sensor. Based on the analysis of the parallel parking process of experienced drivers and the idea of simulating a human driver, the initial path is planned via an arc-line-arc three segment composite curve and fitted by a quintic Bézier curve to make up for the discontinuity of curvature. Then, the radial basis function neural network is established, and slopes of points of the initial path are used as input to train and obtain horizontal ordinates of four control points in the middle of the Bézier curve. Finally, simulation experiments are carried out by MATLAB, whereby parallel parking of driverless vehicle is simulated, and the effects of the proposed method are verified. Results show the trained and optimized Bézier curve as a planning path meets the requirements of continuous curvature, safety, and curvature constraints, thus improving the abilities for parallel parking in small parking spaces.

**Keywords:** radial basis function neural network; path planning; Bézier curve; parallel parking; optimization; driverless vehicle

## 1. Introduction

It is an important working condition for driverless vehicles to complete safe parallel parking in narrow parking spaces, where it is more difficult to park than in ordinary parking conditions [1]. Planning a feasible and safe path with curvature continuity is crucial for autonomous parking yet remains to be fully solved [2,3]. At present, the main methods of path planning in parallel parking are based on geometry, sampling, and numerical optimization. The path planning method based on sampling or numerical optimization can plan a feasible parking path in a complex environment and small parking space but with the disadvantage of a large amount of calculation. The method of path planning based on geometry has high computational efficiency, good real-time performance, and is convenient for engineering implementation [4]. Common parking path curves designed include fifth-order polynomial [5], arc [6], Bézier curve, etc. For example, the path in parallel parking is divided into the entry section to guide the car to approach the parking space and the adjustment section to guide the car to adjust the target pose. The constrained optimization problem of arc-line-cyclotron curve is solved to establish the entry section path, the adjustment section path is planned through the combination of arc-line, guiding the car to complete parking task safely and comfortably [7]. The path planning method of a non-parallel initial state based on global planning to establish a three-arc parking path has stronger robustness than the traditional minimum radius method [8]. In the path planning method based on the improved quintic polynomial and designed penalty function, the

genetic algorithm is used to calculate the optimum parking path and minimum parking space to complete fast and effective parking with less vehicle damage and low space requirements [9].

The path planning problem is often converted to an optimization problem [10,11]. In a guaranteed collision-free trajectory planning method, a dynamic optimization problem is built, including vehicle kinematics, collision avoidance constraints, and physical restraints. To describe collision avoidance constraints, the concepts of the virtual protection frame and the magnification parameter are introduced. Tests in three common scenarios and a complex scenario illustrate the effectiveness [1]. A method based on the Frenet coordinate system is proposed under more complex driving scenarios such as curving roads. To balance the driving comfort ability and safety of the path candidate, a sequential quadratic programming algorithm is employed [12]. A scene-dependent B-spline parallel parking path planning algorithm meets vehicle kinematics model constraints well. Here, the vehicle actuator's dynamic performance and parking path smoothness are key attributes [13]. To solve the multi-vehicle cooperative autonomous parking trajectory planning problem uniformly and effectively with a faster convergence speed, a vehicle kinematics model with dynamic constraints, namely endpoint and collision avoidance constraints, is established. The problem is transformed into an optimal control problem with the shortest parking completion time as the optimal cost function. An initial guess generation strategy is proposed, and the iteration method is used to solve by adding collision avoidance constraints to improve the convergence and robustness [14].

However, the method of integrating a neural network to optimize the comprehensive performances of the initial path planned by the geometric method needs further study.

The rest of the paper is organized as follows. In Section 2, the composition and working process of an autonomous parking system for driverless vehicle are analyzed. Experiments concerning parking space detection based on an Arduino intelligent minicar and ultrasonic sensor are carried out. Based on the idea of simulating a human driver and the geometric method, the parallel parking path of arc-line-arc is initially planned. In order to make up for the discontinuity of curvature of the initial planning path, a Bézier curve is used to fit it. The radial basis function neural network (RBFNN) is established, and the slope of each point of the initial path is used as the input to train and obtain the horizontal ordinates of four control points in the middle of the Bézier curve. In Section 3, based on the MATLAB simulation experiment, the optimized Bézier curve is verified to improve the ability of the driverless vehicle for safer and smoother parallel parking in narrow parking spaces. Conclusions are drawn in Section 4.

## 2. Methods

### 2.1. Composition and Working Process of Autonomous Parking System for Driverless Vehicle

2.1.1. Composition of Autonomous Parking System for Driverless Vehicle

The autonomous parking system of the driverless vehicle consists of the following, as shown in Figure 1.

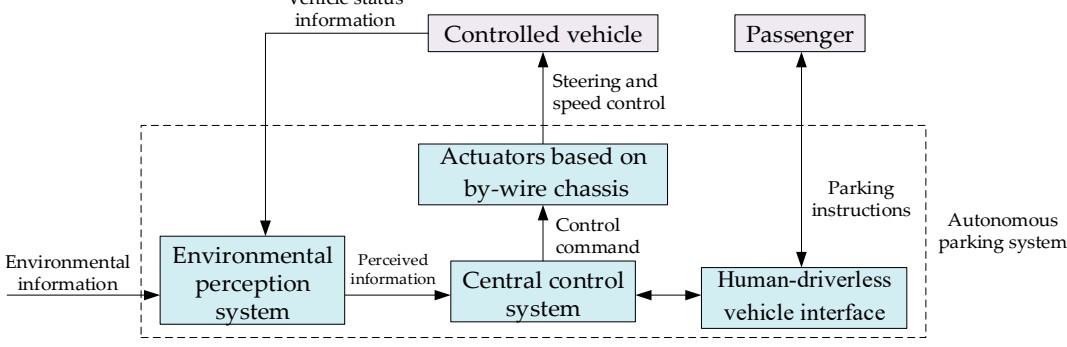

**Figure 1.** Composition of autonomous parking system.

(1) Human-driverless vehicle interface: It is a medium for passengers of driverless vehicle to convey parking intention to the vehicle central control system when they have parking needs. After receiving the instructions, the central control system operates the vehicle to complete parking task.

(2) Environmental perception system: It consists of Lidar, ultrasonic radar, millimeter wave radar, camera, etc., and is responsible for the detection of vehicle driving position information, parking space, and surrounding obstacle information, etc., and input relevant information into the central control system for further decision-making. Precise environmental perception is the premise of smooth autonomous parking.

(3) Central control system: As the core part of autonomous parking system, it processes and analyzes the information from the environmental perception system to judge whether the parking space can meet the parking requirements. On the premise of meeting the parking requirements, according to the real-time attitude and position information of the vehicle body and parking space information, the parking strategy is formulated and control commands, including the front wheel angle, vehicle speed, etc., are output and transmitted to the actuators to realize steering, braking, etc.

(4) Actuators: The actuators based on a by-wire chassis consist of the steering-by-wire system, braking-by-wire system, and driving-by-wire system. They receive the control command information from the central control system in real time, operate the steering-by-wire system, braking-by-wire system, and driving-by-wire system according to the command, and control the front wheel angle and vehicle speed respectively.

2.1.2. Working Process of Autonomous Parallel Parking

The working process of autonomous parallel parking is shown in Figure 2.

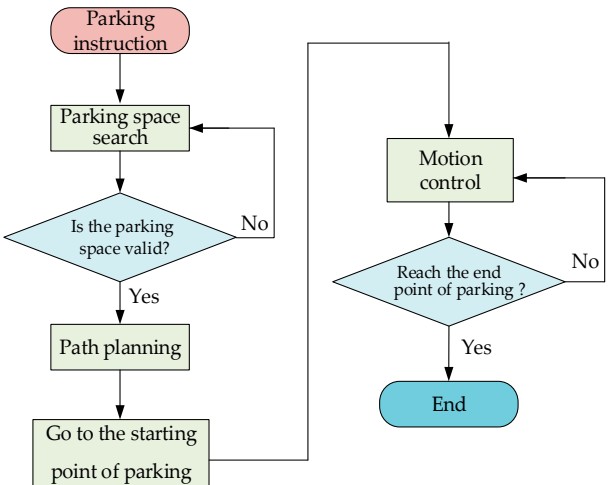

**Figure 2.** The working process of autonomous parking system.

(1) After the vehicle arrives at the destination, the passengers issue parking instructions. The vehicle enables the parking space search function to find free parking space within a certain range of the destination and judge if the found parking space meets the parking conditions until a feasible parking space is found.

(2) After the parking space detection is completed, the autonomous parking system plans a safe and collision free parking path, and the vehicle drives to the parking starting point.

(3) The autonomous parking system controls the movement of the vehicle by outputting the front wheel angle and vehicle speed signals to make it track the planned path. Finally, the vehicle drives into the parking space smoothly and without collision.

### 2.2. Modeling

2.2.1. Vehicle Structure Simplification

In the process of parallel parking, vehicle parameters, such as wheelbase, vehicle length, vehicle width, etc., affect the size of valid parking space and parking path planning. The body structure is composed of irregular surfaces whose analysis is complex. In order to simplify the analysis process and not affect accuracy of the analysis results, the body contour is simplified. The vehicle is regarded as a rigid body moving on a plane and represented by a rectangle. The body is composed of smooth curved surfaces. At the four body vertices, there are arc surfaces, and the top view of the body is not a rectangle. However, according to the analysis of simplified vehicle model shown in Figure 3, the driving area of the vehicle is slightly larger than that of the real vehicle. The collision free path planned will not produce collision when applied to the real vehicle.

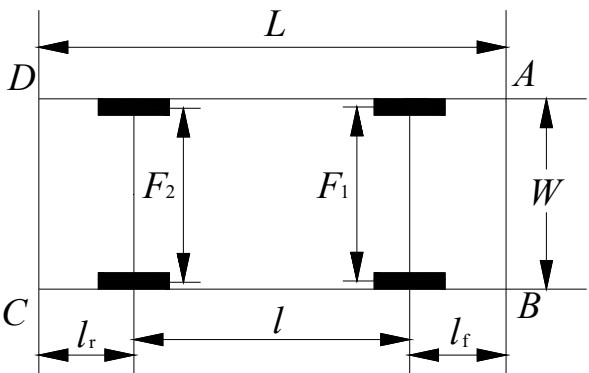

**Figure 3.** Simplified vehicle model for autonomous parallel parking.

In Figure 3, $L$ is the body length, $W$ is the body width, $l$ is the vehicle wheelbase, $l_f$ and $l_r$ are the front and rear overhang respectively, $F_1$ and $F_2$ are the front track and rear track respectively, and $A$, $B$, $C$, and $D$ are the left front vertex, right front vertex, right rear vertex, and left rear vertex of the vehicle respectively.

2.2.2. Vehicle Kinematics Modeling

During steering, all wheels must roll around the same instantaneous rotation center, which is called the Ackerman steering principle, as shown in Figure 4. The front wheels are steered wheels. The angle of the inner and outer front wheel are different.

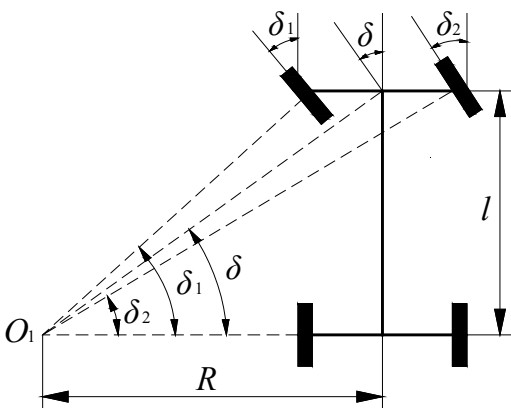

**Figure 4.** Ackerman principle.

In Figure 4, $R$ is the turning radius at the midpoint of the vehicle rear axle, $\delta_1$ and $\delta_2$ are the steering angles of the inner and outer front wheel respectively, $\delta$ is the equivalent

angle at the center of vehicle front axle, and the point $O_1$ is the instantaneous steering center of the vehicle.

From the geometric relationship in Figure 4, it is concluded that there are the following relationships between the inner and outer front wheel angle:

$$\tan \delta = \frac{l}{R} \tag{1}$$

$$\cot \delta_2 - \cot \delta_1 = \frac{F_2}{l} \tag{2}$$

$$\cot \delta_1 + \cot \delta_2 = 2 \cot \delta \tag{3}$$

Through Equation (3), the relationship between the angle of the two front wheels and the equivalent angle at the center of vehicle front axle can be obtained, but there is no intuitive reflection. In order to obtain a clearer relationship, Equations (2) and (3) are used to compare the equivalent angle at the center of vehicle front axle with average value of two front wheel angles by MATLAB. As shown in Figure 5, the gap between the equivalent angle at the center of vehicle front axle and average value of two front wheel angles is very small. Therefore, the following approximation is made:

$$\delta = \frac{\delta_1 + \delta_2}{2} \tag{4}$$

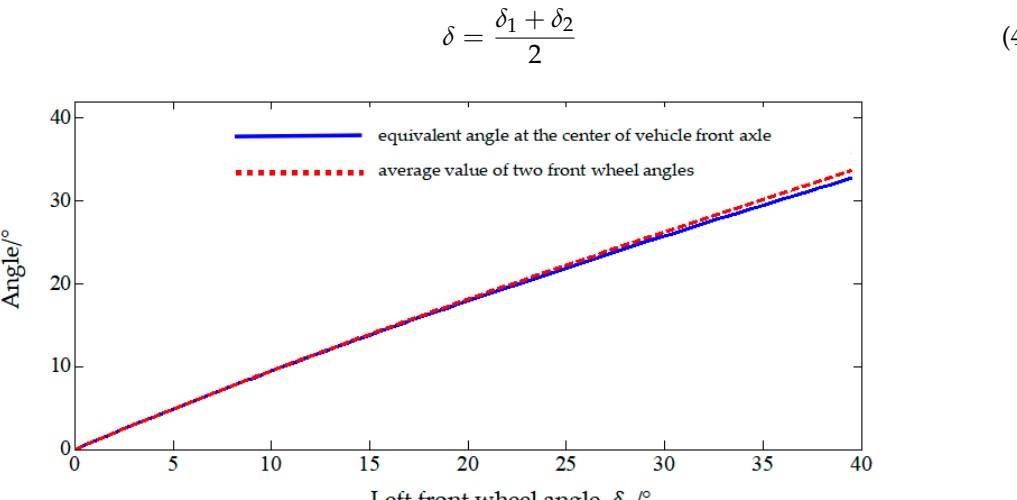

**Figure 5.** Relationship between equivalent angle at the center of vehicle front axle and average value of two front wheel angles.

By finding out the relationship between the front axle center equivalent angle and the front wheel angle, the difficulty of modeling is reduced. In the follow-up study, the equivalent angle at the center of the vehicle front axle is used to replace the front wheel angle for modeling.

### 2.2.3. Schematic Diagram of Parallel Parking Condition

Figure 6 presents a schematic diagram of the parallel parking condition, showing the parking space with large traffic flow and tight parking space, and there are vehicles in front of and behind the target parking spaces.

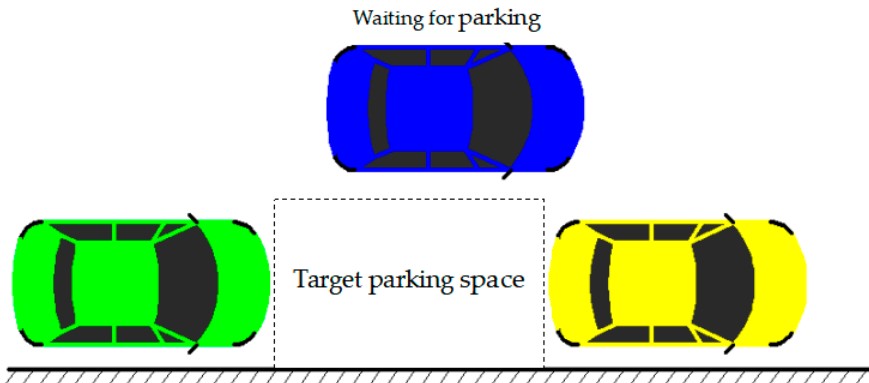

**Figure 6.** Schematic diagram of parallel parking condition.

*2.3. Experiments of Parking Space Detection Based on Ultrasonic Sensor*

2.3.1. The Detection Process of the Size of Parking Space and Experimental Equipment

The size of parking space can be calculated in combination with the return value of the ultrasonic sensor shown in Figure 7.

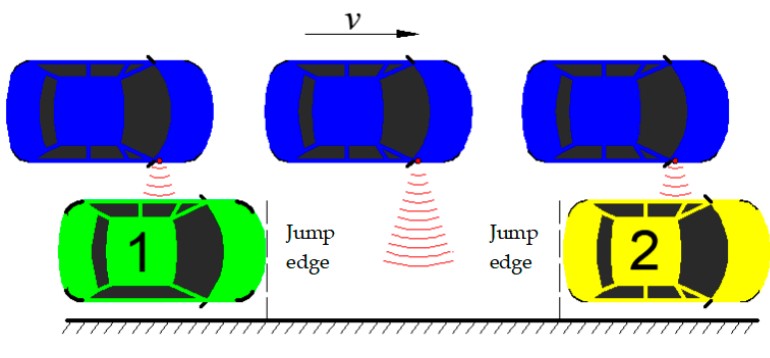

**Figure 7.** The detection process of the size of parking space.

(1) The target vehicle (in blue) runs in a straight line at a low speed. When passing vehicle 1, the ultrasonic sensor on the side of the vehicle starts to detect the distance between the two vehicles. Due to the irregularity of the body surface, the detection value obtained fluctuates in a small range.

(2) When the target vehicle gradually drives away from vehicle 1, the ultrasonic sensor loses the vehicle that can be detected, and the detection value will suddenly increase, resulting in an obvious jump at the front edge of vehicle 1. As the vehicle travels, when the sensor detects vehicle 2, the detection value suddenly decreases, an obvious jump occurs at the rear edge of vehicle 2, and the vehicle travel distance S between the two jumps of the detection value is recorded. After that, the vehicle continues to move forward, and the ultrasonic sensor obtains a detection value with small fluctuation.

(3) After parking space detection is completed, the driving distance $S$ is compared with preset length $L$ of the minimum parking space required for parking. If $S \geq L$, the path will be planned to prepare for parking, whereas if $S < L$, it continues to find suitable parking spaces.

In order to verify the feasibility of parking space detection method by ultrasonic, an intelligent minicar driven by the rear wheels and controlled by Arduino single chip microcomputer is used as the experimental equipment, as shown in Figure 8. The driving motor is a DC motor of model MG5130P30 with Hall encoder, which can realize the speed measurement function. The rated voltage of the motor is 12 V and the rated power is

7 W. The car is equipped with a Bluetooth module BT04, which can wirelessly return the detected distance data through serial port monitor of Arduino.

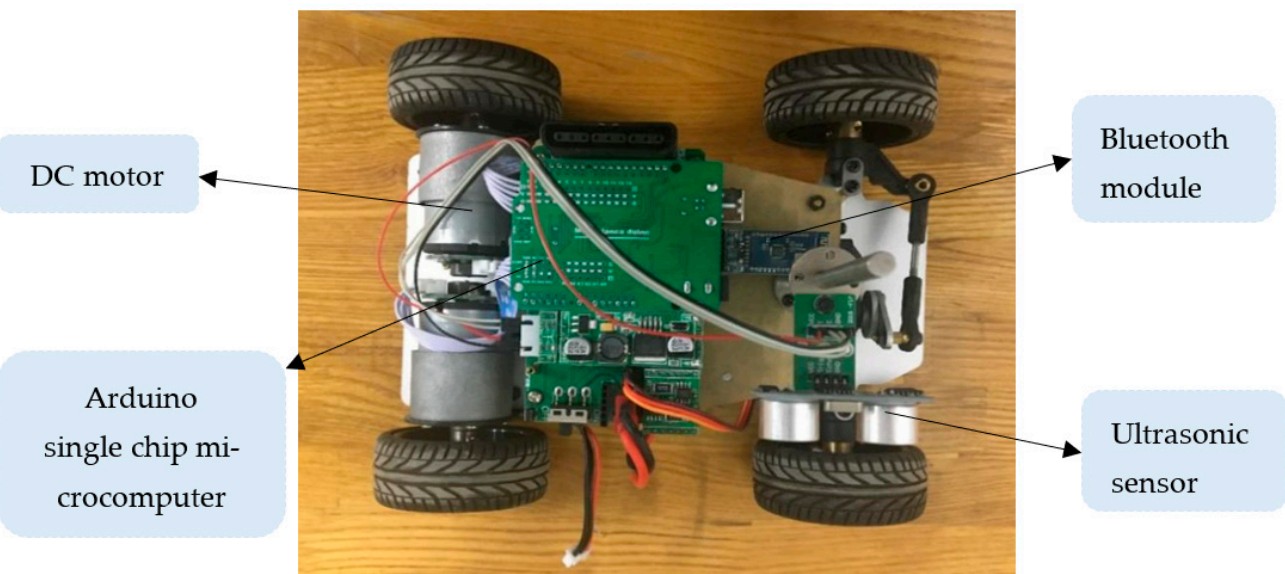

**Figure 8.** Experimental equipment—Arduino minicar.

2.3.2. Experiments of Parking Space Detection Based on Ultrasonic Sensor

Ultrasonic sensor is a distance sensor based on the principle of ultrasonic reflection. The distance between the parked vehicles and the body of ego vehicle is small, so the advantages of ultrasonic sensors are fully exerted. The echo is formed by ultrasonic reflection when it touches obstacles. The transit time detection method is used to measure the distance by the parameter difference between the transmitting wave and the return wave. The intelligent minicar controlled by Arduino microcontroller is used as the experimental equipment. The simulated parking space with length of 1 m and width of 0.52 m is set up and a simulation experiment scene is built according to the parallel parking conditions. Objects ① and ② with certain volume and shape are used to simulate the parked vehicles 1 and 2, and object ③ is used to represent the road boundary. The experimental scene is shown in Figure 9a. The motor speed is fixed by analogWrite function, and the analog of input pin is 40 (square wave with duty cycle of 40/255 is output by PWM) to make the speed in each experiment consistent.

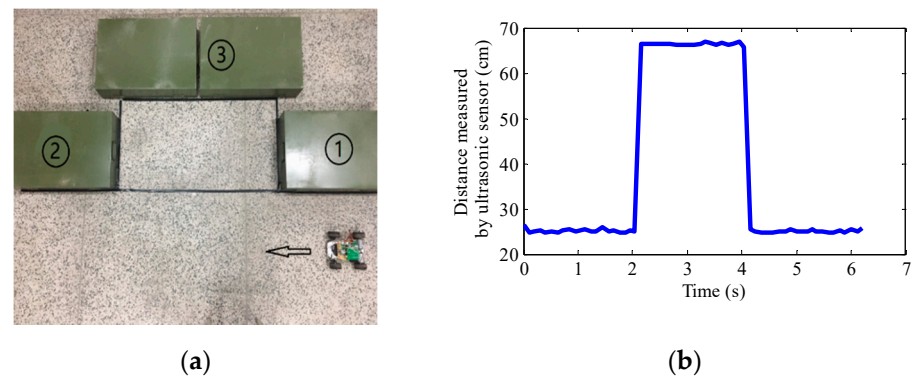

(a)  (b)

**Figure 9.** Parking space detection experiment for parallel parking: (**a**) Experiment scene; (**b**) Experimental results of a parking space detection.

At the beginning of the experiment, each time the car starts from the same position, the lateral distance between the ultrasonic sensor and object ① is 25 cm. It runs straight at the set speed through the simulated parking space. In order to eliminate the chance of the experiment, 21 experiments are carried out, and the data collected by the ultrasonic sensor are analyzed. In Figure 9b, the data used in one experiment are selected for illustration. From the distance measured by the ultrasonic sensor-time curve, the experimental results are in line with expectations. In 0~2.16 s, the car passes through object ①, and the distance between the car and the object ① is measured by the ultrasonic sensor. The ranging value fluctuates in a minimum range of 25 cm within the allowable error range, basically it changes into a straight line with time. When the time is 2.16 s, the ultrasonic sensor of the minicar just reaches the edge of the object ① and the ranging value changes suddenly. The return value of the ultrasonic sensor changes into the distance between the sensor and the object ③. The ranging value changes in a small range of 67 cm from 2.16 s to 4.04 s. At 4.04 s, the ultrasonic sensor reaches the edge of the object ②, and the ranging value changes into the distance between the sensor and object ②, then mutates again, and changes within a minimum range of 25 cm until it leaves the object ②. The fluctuation of ranging value is caused by the small shaking of the minicar in the process of driving. The measured value of parking space width is basically consistent with the true value and fluctuates around the true value.

The experimental results show that the length of the parking space can be calculated by combining the time difference between two jumps of the ultrasonic sensor ranging value under the condition of obtaining vehicle speed, and the width of the parking space can be measured by the ultrasonic sensor ranging value.

### 2.4. Initial Path Planning of Parallel Parking

#### 2.4.1. Analysis of Experienced Driver's Parking Process

The parallel parking process of experienced drivers when there are vehicles in front and behind the waiting parking space is analyzed as shown in Figure 10. First of all, the driver selects appropriate starting position of parking, shifts into reverse gear, turns the steering wheel to the right limit position, and the vehicle runs along the arc. When he observes that the rear of the vehicle is close to the right side line of the parking space, he quickly turns the steering wheel to the left limit position and continues to reverse. When both sides of the vehicle body are parallel to the side line of the parking space, he quickly turns the steering wheel back to the right, stops, and the parking is over. The vehicle path is similar to the S-curve. The parallel parking mode is reversing. Combined with the Ackerman steering principle, taking the midpoint of the rear axle as the reference point, from which the trajectory of any point of the body is obtained, the parking path of the vehicle is analyzed.

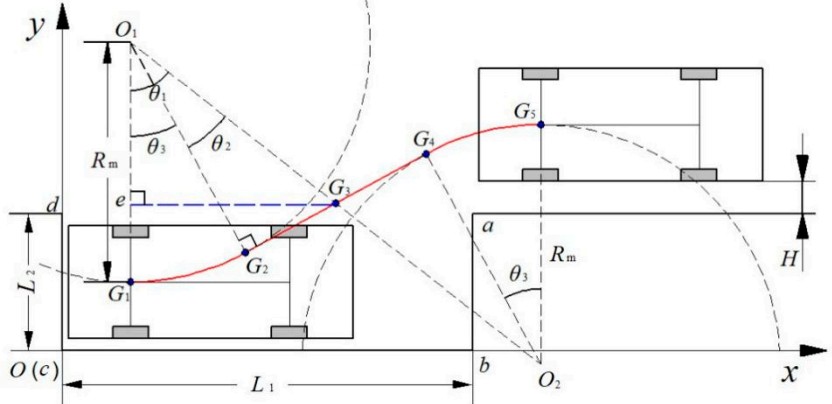

**Figure 10.** Arc- line-arc parallel parking path.

The conditions for planning the parking path include:

(1)　Safety, i.e., the vehicle will not collide with the surrounding obstacles when parking according to the planning path so as to ensure the safety of human and vehicles.

(2)　Meeting the constraints of path curvature, i.e., the minimum curvature radius of the path is greater than the minimum turning radius of the vehicle so as to track the path.

(3)　Curvature continuity, i.e., the curvature of the parking path is continuous everywhere, because continuous curvature can ensure the continuous steering control of the vehicle and avoid the situation of pivot steering.

### 2.4.2. Initial Path Planning of Arc-Line-Arc Based on Idea of Simulating Human to Drive

Based on the idea of simulating a human driver [15,16] and the parallel parking process of experienced drivers, the path planning is carried out. The vehicle path is composed of two arcs. When the vehicle moves to middle position, the steered wheels are turned from the right limit position to the left limit position. Vehicle pivot steering will aggravate tire wear and driving with the minimum steering radius for a long time will cause a certain burden on the steering system. In order to reduce the vehicle loss, a straight line transition is introduced between the two arcs, so the parking path is planned based on the arc-line-arc three segment combined curve.

As shown in Figure 10, the rectangular coordinate system is established to plan the parking path, the right rear corner $O(c)$ of the parking space is taken as the coordinate origin, the direction $Ob$ parallel to the road is $x$ axis, and the direction $Od$ perpendicular to the road is $y$ axis with the length unit of m and the angle unit of rad. $G_5$ is the starting point of parking, $G_1$ is the end point of parking. The planning path consists of arc section $G_5G_4$, straight line section $G_4G_2$ and arc section $G_2G_1$. $G_4G_2$ is tangent to $G_5G_4$ and $G_2G_1$. $G_3$ is the midpoint of the parking path. The point $O_2$ is the center of the trajectory circle where $G_5$ is located, and $O_1$ is the center of the circle in which $G_1$ is located. The steering wheel is cancelled in the driverless vehicle with by-wire chassis and the steered wheels are turned by the steering motor assembly directly in steering -by-wire system. The parking process includes three stages: the steered wheels are turned to the right to the limit position at $G_5$, reverse the steering to point $G_4$ with the minimum turning radius $R_m$; return to the normal direction, reverse to point $G_2$ in straight line; the steered wheels are turned to the limit position to the left, reverse steering to point $G_1$, and return to the right position to complete parking.

The coordinates of $G_1$ and $G_5$ are set respectively as $(x_1, y_1)$ and $(x_5, y_5)$, so the coordinates of $G_3$ are

$$[x_3, y_3] = [(x_1 + x_5)/2, (y_1 + y_5)/2] \tag{5}$$

In Figure 10, a vertical line is made of line segment $O_1G_1$ through point $G_3$ and intersect line segment $O_1G_1$ at point $e$. In the right triangle $O_1eG_3$, the following relationship is established.

$$\sin\theta_1 = \frac{eG_3}{O_1G_3} = \frac{x_3 - x_1}{\sqrt{(x_3 - x_1)^2 + (y_1 + R_m - y_3)^2}}$$

So, the angle $\theta_1$ can be derived as

$$\theta_1 = \arcsin\frac{x_3 - x_1}{\sqrt{(x_3 - x_1)^2 + (y_1 + R_m - y_3)^2}} \tag{6}$$

Similarly, in the right triangle $O_1G_2G_3$, Equation (7) can be derived according to the following relationship

$$\cos\theta_2 = \frac{O_1G_2}{O_1G_3} = \frac{R_m}{\sqrt{(x_3 - x_1)^2 + (y_1 + R_m - y_3)^2}}$$

$$\theta_2 = \arccos\frac{R_m}{\sqrt{(x_3 - x_1)^2 + (y_1 + R_m - y_3)^2}} \tag{7}$$

$$\theta_3 = \theta_1 - \theta_2 \tag{8}$$

Through coordinates of point $G_1$, $G_5$ and value of $\theta_3$, coordinates of point $G_2$ (Formula (9)) and coordinates of point $G_4$ (Formula (10)) are obtained:

$$[x_2, y_2] = [x_1 + R_m \sin\theta_3, y_1 + R_m(1 - \cos\theta_3)] \tag{9}$$

$$[x_4, y_4] = [x_5 - R_m \sin\theta_3, y_5 - R_m(1 - \cos\theta_3)] \tag{10}$$

### 2.4.3. Determination of the Starting and Ending Points of Parking

According to the requirements of collision constraints, the vehicle is most likely to collide with the parking space vertex $a$ in the parking process, so the right contour of the vehicle body should be at the upper left of point $a$ when the vehicle is in $G_2G_3$ segment. The vehicle should keep a safe lateral distance $H$ from the vehicle ahead at $G_5$ the starting point of parking.

The limit situation of vehicle collision free is analyzed in Figure 11. When the vehicle enters the straight line section, the distance between point $a$ and the straight section is greater than half of the width of the body $W$, and the circle centered on point $O_2$ and the circle centered on point $a$ are tangent to $G_7$. $\theta_a$, $\theta_b$, and $\theta_{3min}$, the minimum center angle of arc $G_2G_1$, are obtained from the coordinates of point $G_1$ $(x_1, y_1)$, coordinates of point $a$ $(L_1, L_2)$.

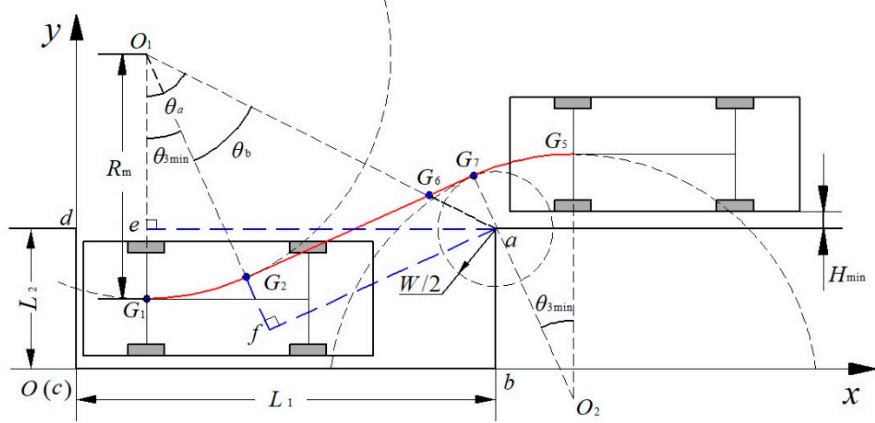

**Figure 11.** Extreme position of parallel parking with $\theta_{3min}$.

In Figure 11, a vertical line is made of line segment $O_1G_1$ through point $a$ and intersect line segment $O_1G_1$ at point $e$. In the right triangle $O_1ea$,

$$\sin\theta_a = \frac{ea}{O_1a} = \frac{L_1 - x_1}{\sqrt{(L_1 - x_1)^2 + (y_1 + R_m - L_2)^2}}.$$

Equation (11) can be derived as

$$\theta_a = \arcsin\frac{L_1 - x_1}{\sqrt{(L_1 - x_1)^2 + (y_1 + R_m - L_2)^2}} \tag{11}$$

Similarly, in the right triangle $O_1fa$, Formula (12) can be derived according to

$$\cos\theta_b = \frac{O_1f}{O_1a} = \frac{R_m + W/2}{\sqrt{(L_1 - x_1)^2 + (y_1 + R_m - L_2)^2}}$$

$$\theta_b = \arccos \frac{R_m + W/2}{\sqrt{(L_1 - x_1)^2 + (y_1 + R_m - L_2)^2}} \tag{12}$$

$$\theta_{3min} = \theta_a - \theta_b \tag{13}$$

From the above angle values, the coordinates of point $G_2$ (Equation (14)), point $G_6$ (Equation (15)), and point $G_7$ (Equation (16)) can be obtained:

$$[x_2, y_2] = [x_1 + R_m\sin\theta_{3min}, y_1 + R_m(1 - \cos\theta_{3min})] \tag{14}$$

$$[x_6, y_6] = [x_2 + R_m\tan\theta_b\cos\theta_{3min}, y_2 + R_m\tan\theta_b\sin\theta_{3min}] \tag{15}$$

$$[x_7, y_7] = [x_2 + (R_m + W/2)\tan\theta_b\cos\theta_{3min}, y_2 + (R_m + W/2)\tan\theta_b\sin\theta_{3min}] \tag{16}$$

The coordinates of parking starting point $G_5$ are

$$[x_5, y_5] = [x_7 + R_m\sin\theta_{3min}, y_7 + R_m(1 - \cos\theta_{3min})] \tag{17}$$

The minimum lateral safe distance at the starting point of parking is

$$H_{min} = y_5 - W/2 - L_2 \tag{18}$$

By substituting the vehicle structure parameters into the calculation formula of the minimum parking space size, the minimum parking space length $L_1$ = 5.89 m and the minimum parking space width $L_2$ = 1.74 m are obtained. Considering the safety distance of anti-collision, the minimum parking space size is adjusted to $L_1$ = 6.5 m, $L_2$ = 2 m. The rear overhang of the vehicle is 0.905 m. Considering the collision avoidance requirements and parking standardization, the coordinates of $G_1$ the parking end point is set as (1 m, 1 m). According to Equation (18), $H_{min}$ = 0.24 m, which is adjusted to 0.3 m to ensure the safety of parking path. According to the parking experience, when the rear of the ego vehicle is flush with the rear of the side vehicle, it starts to steer, and finally the coordinates of $G_5$ the parking starting point are determined as (7.5 m, 3.1 m).

In order to discuss the possible maximum value of angle $\theta_3$ i.e., $\theta_{3max}$ or the maximum distance between point $G_7$ and point $G_5$, extreme position of parallel parking showing $\theta_{3max}$ is illustrated in Figure 12. The value of angle $\theta_3$ is related to the position of point $G_4$. If the vehicle parks with the minimum turning radius, regardless of the road width limit, when the length of the straight line segment $O_1O_2$ connecting the two arcs is 0 (the two arcs are directly tangent) and the straight line segment $O_1O_2$ is parallel to the lane line, the maximum value of angle $\theta_3$ can be obtained. At this time, point $G_2$, $G_3$, and $G_4$ coincide to the same point and the maximum value of angle $\theta_3$ is 90°.

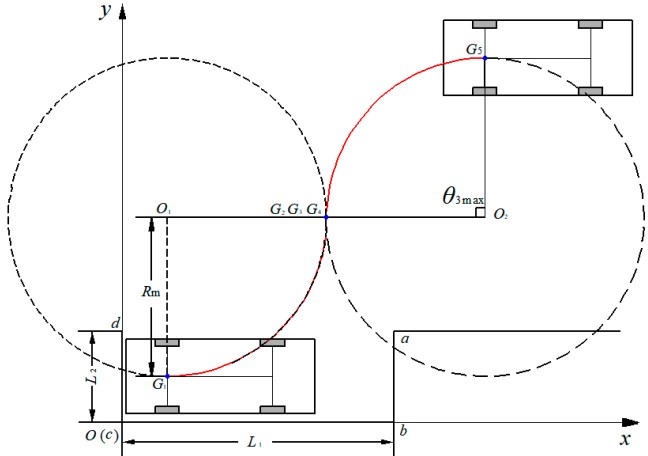

**Figure 12.** Extreme position of parallel parking showing $\theta_{3max}$.

If the vehicle does not drive with the minimum turning radius in section $G_5G_7$, the arc length of arc $G_5G_7$ will become larger. The larger the turning radius, the longer the arc length of the corresponding arc $G_5G_7$, which depends on the maximum turning radius of the vehicle. Since point $G_7$ is a definite point, no matter what the steering radius is, the value of angle $\theta_3$ will not change at this time.

2.4.4. Characteristics of Initial Planning Path of Arc-Line-Arc

The path of arc-line-arc three segment combined curve meets the parking safety and curvature constraints. However, as shown in Figure 13, the curvature of the whole path is not continuous, and there is obvious curvature mutation at the intersection of arc and line, because the vehicle needs pivot steering at these points. In order to make the curvature of the path meet the condition of curvature continuity, the initial path is fitted by the Bézier curve and optimized by the radial basis function neural network method.

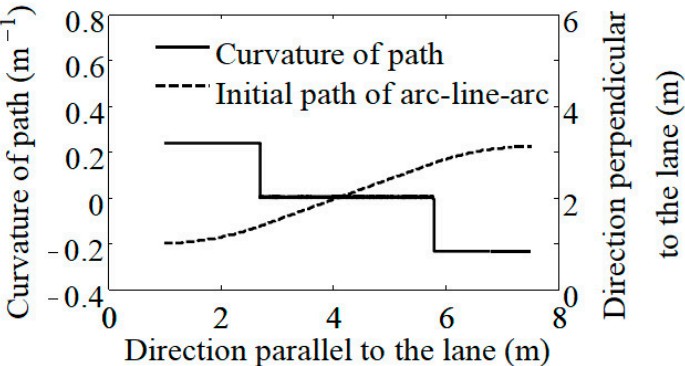

**Figure 13.** Curvature change of arc-line-arc path.

*2.5. Path Optimization Integrating Bézier Curve and Radial Basis Function Neural Network*
2.5.1. Bézier Curve

Bézier curve is a smooth and continuous engineering curve, which can describe complex shape and show curve trend through control points. Through analysis, the Bézier curve needs at least six control points to satisfy curvature continuity constraint, that is, at least fifth order. Because the higher the degree of the Bézier curve, the more complex the curve [17], the fifth order Bézier curve shown in Equation (19) is selected to fit the initial planning path of arc-line-arc.

$$P(u) = (1-u)^5P_0 + 5(1-u)^4 u \, P_1 + 10(1-u)^3 u^2P_2 + 10(1-u)^2 u^3P_3 + 5(1-u) u^4P_4 + u^5P_5 \qquad (19)$$

where $u$ is the parameter of the Bézier curve, $u \in [0, 1]$; $P_0$, $P_1$, $P_2$, $P_3$, $P_4$, and $P_5$ are control points of the quintic Bézier curve.

When the vertical coordinates of the first three control points $P_0$, $P_1$ and $P_2$ are the same, and the vertical coordinates of the last three control points $P_3$, $P_4$ and $P_5$ are also the same, the Bézier curve is S-shaped, which is similar to the parallel parking curve. The parking end point $G_1$ and the parking starting point $G_5$ are regarded as the first control point $P_0$ and the sixth control point $P_5$ of the Bézier curve respectively.

A schematic diagram of quintic Bézier curve with six control points $P_0$, $P_1$, $P_2$, $P_3$, $P_4$, and $P_5$ is shown in Figure 14. The values of horizontal ordinates and vertical coordinates of the parking end point $G_1$ and the parking starting point $G_5$ have been determined in Section 2.4.3 as $G_1$ (1, 1) and $G_5(7.5, 3.1)$ separately. Accordingly, the values of horizontal ordinates and vertical coordinates of the point $P_0$ and $P_5$ of the quintic Bézier curve are known as $P_0(1, 1)$, $P_5(7.5, 3.1)$. Then, the vertical coordinate of the control point $P_1$ and $P_2$ is equal to 1 m, and the vertical coordinate of the point $P_3$ and $P_4$ is equal to 3.1 m. Only the horizontal ordinates of the four control points $P_1$, $P_2$, $P_3$, and $P_4$ in the middle of the quintic Bézier curve are unknown and need to be trained by the following radial basis function neural network.

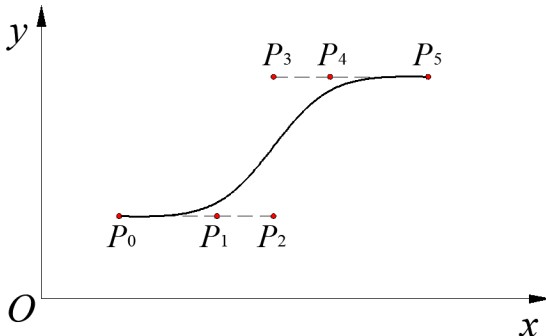

**Figure 14.** Schematic diagram of quintic Bézier curve.

2.5.2. Radial Basis Function Neural Network

When the coordinates of control points are known, it is easy to represent the shape of the Bézier curve. However, it is very difficult to obtain the coordinates of control points by the curve shape, which can be solved by the strong learning ability of neural network. The radial basis function is applied to the design of the neural network and forms the radial basis function neural network, which has good nonlinear approximation ability, fast training speed, and no local optimal problem. It is a three-layer forward network, with only one hidden layer. The first layer is input layer, which is composed of signal source nodes. The second layer is the hidden layer, the transformation function of the hidden element is radial basis function, and the number of hidden elements is determined by the need of solving the problem. The third layer is the output layer, and the output layer node is a simple linear function. The output of the hidden layer unit is weighted linearly to get the network output. The transformation between input layer and hidden layer is nonlinear, and the transformation from hidden layer to output layer is linear. As shown in Figure 15, the established radial basis function neural network has 101 input nodes, taking the slope of 101 points in the initial path; $n$ hidden layer units; 4 output layer nodes, namely the horizontal ordinates of the four control points in middle of the fifth order Bézier curve. Network input $x = [x_1, x_2, x_3, \ldots, x_{101}]$, network output is $y = [y_1, y_2, y_3, y_4]$.

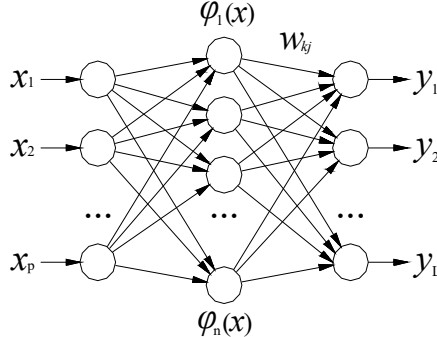

Input layer    Hidden layer   Output layer

**Figure 15.** The structure of radial basis function neural network.

2.5.3. Calculation Process of Radial Basis Function Neural Network

The input $x$ of the neural network shown in Figure 15 needs to be transformed through radial basis function of the hidden layer to obtain the output of hidden unit. Then, the output of the hidden unit is linearly weighted and summed to obtain the output $y$ of the network. Among many kinds of radial basis functions in common use, the Gaussian function is chosen as the basis function here due to the advantages of simple form, good

analytical property, radial symmetry about the center point, and differentiability of any order with the expression as

$$\varphi_i(x) = e^{-\frac{(x-c_i)^{\mathrm{T}}(x-c_i)}{2\sigma_i^2}} \qquad i = 1, 2, \cdots, n \tag{20}$$

where $\varphi_i$ is the output of the $i$th hidden unit; $c_i$ is the center vector of the Gaussian function of the $i$th hidden element, whose dimension is the same as the input vector $x$; $\sigma_i$ is the expansion constant of the $i$th hidden element, reflecting the flatness of the function; $n$ is the number of hidden layer units.

The outputs of all hidden layer elements are linearly weighted and summed to obtain the output of each node of the output layer as

$$y_k = \sum_{i=1}^{n} w_{ki}\varphi_i - \theta_k \quad k = 1, 2, \cdots, L \tag{21}$$

where $y_k$ is the output of the $k$-th node of the output layer; $\omega_{ki}$ is the weight from the $i$th node of the hidden layer to the $k$th node of the output layer; $\theta_k$ is the threshold of the hidden layer.

### 2.5.4. Construction of Radial Basis Function Neural Network in MATLAB

A strict radial basis function neural network is created by newrb function in MATLAB software. Its characteristic is that the number of hidden units is not determined first, but the hidden units are added to the hidden layer according to the error in the training process until the error meets the requirements. The expansion speed needs to be set. After debugging, its value is set to 2.4.

In order to make the Bézier curve approach arc-line-arc path, the radial basis function neural network is trained. Slope values of 150 groups of Bézier curves are collected as the input of the training samples of the neural network. Each input vector has 101 dimensions, i.e., each Bezier curve is divided into 101 points for slope collection, and the horizontal ordinates of the four control points $P_1$, $P_2$, $P_3$, and $P_4$ in middle of each curve are used as the training output.

Figure 16 shows the neural network after training, with five hidden layer units. Taking the slope data of the arc-line-arc path shown in Figure 12 as the input of the neural network, the horizontal ordinates values of control points $P_1$, $P_2$, $P_3$, and $P_4$ are 2.6442, 1.1935, 6.8065, and 5.3558, respectively.

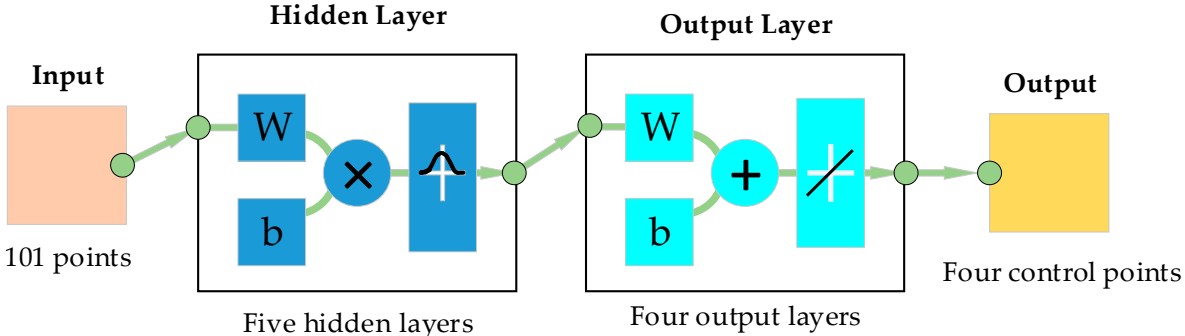

**Figure 16.** Trained radial basis function neural network.

## 3. Simulation Experiment Results

The radial basis function neural network is trained by MATLAB, and the horizontal ordinates of four control points in middle of the fifth order Bézier curve are obtained. After optimization, the coordinates of the six control points of the quintic Bézier curve in

Equation (19) are known as $P_0(1, 1)$, $P_1(2.6442, 1)$, $P_2(1.1935, 1)$, $P_3(6.8065, 3.1)$, $P_4(5.3558, 3.1)$, and $P_5(7.5, 3.1)$ separately.

By substituting six control points into Equation (19), the Bézier curve expression of the optimized path is obtained. Figure 17 shows the performance of Bézier curve path trained by radial basis function neural network. In Figure 17a, the path of the Bézier curve trained by the radial basis function neural network is shown as the solid line. The curvature of the optimized parking path is shown as the dotted line. The curvature curve of the whole path is a smooth and continuous curve, which meets the curvature continuous condition of the parking path. Compared with Figure 13, the curvature of the optimized path changes more gently in the parking process, without obvious mutation, so it is more advantageous to continuously change the front wheel steering angle of the vehicle and reduce the control difficulty.

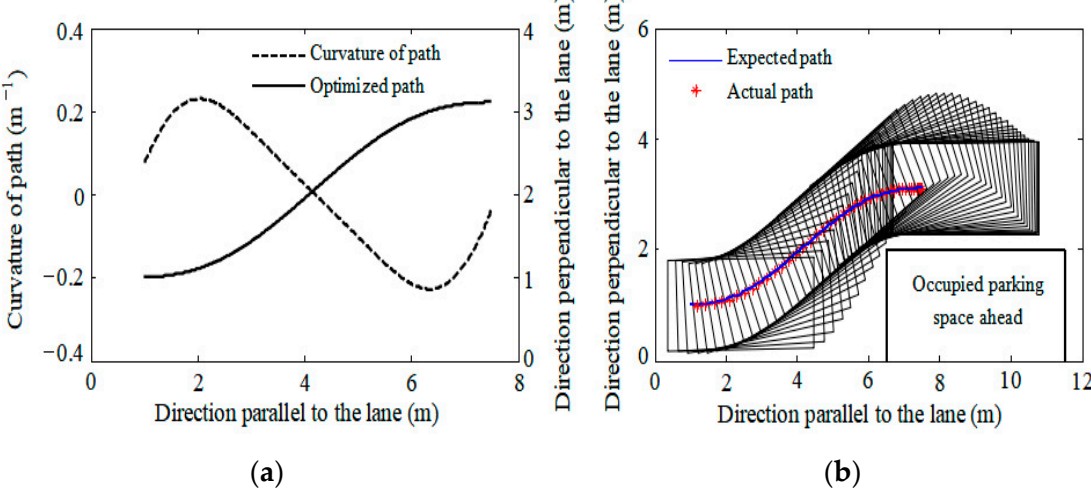

**Figure 17.** Performance of Bézier curve path trained by radial basis function neural network: (**a**) Path of Bézier curve and curvature of path trained by radical basis function neural network; (**b**) Safety of optimized parking path.

The curvature of a point on the arc is the reciprocal of the radius, so the curvature value $|K|$ of arc corresponding to the minimum turning radius $R_m$ is calculated as follows

$$|K| = 1/R_m = 0.2381 \text{ m}^{-1}$$

The maximum curvature value of parallel parking path based on Bézier curve $|K|_{max}$ is 0.2284 m$^{-1}$, which is less than the allowable maximum curvature value 0.2381 m$^{-1}$. So, the path curvature constraint is met.

MATLAB platform is used to model and simulate parallel parking condition. Pure pursuit control algorithm is used to track planning path [10] and verify the safety of the path. During parallel parking, the right rear corner and the right front corner of the vehicle are most likely to collide with the left rear corner of the vehicle in front. As shown in Figure 17b, in the parking process of the tracking planning path, the two points that are most prone to collision do not collide avoiding collision with the vehicles in front. After parking, the rear of the vehicle has a certain distance from the vehicles in the rear, so it does not collide with the vehicles in the rear. Therefore, the planning path optimization meets the parking safety requirements.

## 4. Conclusions

Based on the idea of simulating a human driver and the characteristics of the parallel parking path of skilled drivers, the initial planning path of arc-line-arc is selected. The Bézier curve is used to fit the initial planning path to meet the curvature continuity condition. Through the comparison of path curve curvature changes in Figures 13 and 17a, it can be found that performance of curvature continuity of the Bézier curve path trained

by the radial basis function neural network is better than that of the initial planning path of arc-line-arc. This shows the reasonable optimization efficiency. It makes full use of the advantages of a smooth and continuous Bézier curve, demonstrating good nonlinear approximation ability and fast training of the radial basis function neural network. The path planning meets the requirements of curvature continuity, safety, and curvature constraints of parallel parking for driverless vehicles. Compared with the path planning optimization results of similar research conditions in other literature, e.g., Figure 10 in [18], it reflects the optimization efficiency of the proposed research method in this manuscript insofar as the smooth and continuous advantages of the Bézier curve, good nonlinear approximation ability, and fast training of the radial basis function neural network are concerned.

In future research, a real vehicle test based on the Baidu Apollo D-KIT lite driverless development kit will be carried out to verify the proposed method.

**Author Contributions:** All the authors made significant contributions to this work. Conceptualization, L.Y. and X.W.; methodology, X.W.; software, X.W.; validation, Z.M.; formal analysis, Y.Z.; investigation, Z.D.; resources, Z.H.; data curation, Z.H.; writing—original draft preparation, X.W.; writing—review and editing, Z.H.; visualization, X.W.; supervision, L.Y.; project administration, L.Y. All authors have read and agreed to the published version of the manuscript.

**Funding:** This research was funded by Fundamental Research Funds for the Central Universities, grant number 19CX02019A and Graduate Education and Teaching Reform Project of China University of Petroleum (East China), grant number YJG2019025.

**Institutional Review Board Statement:** Not applicable.

**Informed Consent Statement:** Not applicable.

**Data Availability Statement:** Data available on request due to restrictions of privacy.

**Acknowledgments:** We are thankful to anonymous editor and reviewers for their valuable comments and kind suggestions in early versions of this article.

**Conflicts of Interest:** The authors declare no conflict of interest.

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
