# Peer review of "Path Planning Optimization for Driverless Vehicle in Parallel Parking Integrating Radial Basis Function Neural Network"

_applsci, doi:10.3390/app11178178_

Round 1
Reviewer 1 Report
Interesting article.
A fairly detailed mathematical model of parallel parking is presented, but I missed the analysis, what is the possible Θ3max or the maximum distance between points G7G5?
I also lacked a set of optimal parameters for the simulated situation.
In addition, it would be interesting to see the optimization efficiency, that is, a comparison of the modeling results with the results of solutions to a similar problem by other authors. This could be reflected in the conclusions.
Misunderstanding "Rmin is 1 / Rmin = 0.2381" (lines 396 and 397).
Author Response
Dear editors and referee:
We deeply appreciate your valuable comments and suggestions for our manuscript and have carefully revised our manuscript. The main revisions are as follows.
Please see the attachment.

Reviewer 2 Report
With the paper titled “Path Planning Optimization for Driverless Vehicle in Parallel Parking Integrating Radial Basis Function Neural Network,” the authors investigated a novel method to train control points of Bézier curve using radial basis function neural network method for parallel parking for driverless vehicles. The numeral activity was conducted with MATLAB software, for verifying the effects of the proposed method. The authors show that the trained and optimized Bézier curve meets the requirements of continuous curvature, safety and curvature constraints, improving the abilities of parallel parking for small parking spaces.
The paper is well-written, concise, and easy to understand but it would be interesting to have more information about the numerical activity
Author Response

(The authors gave the same response as above.)
